# Phenol–Hyaluronic Acid Conjugates: Correlation of Oxidative Crosslinking Pathway and Adhesiveness

**DOI:** 10.3390/polym13183130

**Published:** 2021-09-16

**Authors:** Jungwoo Kim, Sumin Kim, Donghee Son, Mikyung Shin

**Affiliations:** 1Department of Biomedical Engineering, Sungkyunkwan University (SKKU), Suwon 16419, Korea; wjddn1998@naver.com (J.K.); ally0618@naver.com (S.K.); 2Department of Intelligent Precision Healthcare Convergence, Sungkyunkwan University (SKKU), Suwon 16419, Korea; 3Department of Electrical and Computer Engineering, Sungkyunkwan University (SKKU), Suwon 16419, Korea; 4Department of Superintelligence Engineering, Sungkyunkwan University (SKKU), Suwon 16419, Korea; 5Center for Neuroscience Imaging Research, Institute for Basic Science (IBS), Suwon 16419, Korea

**Keywords:** hyaluronic acid, phenol, adhesive hydrogels

## Abstract

Hyaluronic acid (HA) is a natural polysaccharide with great biocompatibility for a variety of biomedical applications, such as tissue scaffolds, dermal fillers, and drug-delivery carriers. Despite the medical impact of HA, its poor adhesiveness and short-term in vivo stability limit its therapeutic efficacy. To overcome these shortcomings, a versatile modification strategy for the HA backbone has been developed. This strategy involves tethering phenol moieties on HA to provide both robust adhesiveness and intermolecular cohesion and can be used for oxidative crosslinking of the polymeric chain. However, a lack of knowledge still exists regarding the interchangeable phenolic adhesion and cohesion depending on the type of oxidizing agent used. Here, we reveal the correlation between phenolic adhesion and cohesion upon gelation of two different HA–phenol conjugates, HA–tyramine and HA–catechol, depending on the oxidant. For covalent/non-covalent crosslinking of HA, oxidizing agents, horseradish peroxidase/hydrogen peroxide, chemical oxidants (e.g., base, sodium periodate), and metal ions, were utilized. As a result, HA–catechol showed stronger adhesion properties, whereas HA–tyramine showed higher cohesion properties. In addition, covalent bonds allowed better adhesion compared to that of non-covalent bonds. Our findings are promising for designing adhesive and mechanically robust biomaterials based on phenol chemistry.

## 1. Introduction

Hyaluronic acid (HA) is a natural polysaccharide constructed from two alternating units of N-acetyl-D-glucosamine and D-glucuronic acid [1,2]. It is an important component of the extracellular matrix and plays a role in wound healing and in controlling the release of growth factors [3,4,5]. Previous research has further shown that HA is very versatile in its use in medical treatment and tissue engineering because of its high biocompatibility, biodegradability, viscoelasticity, and non-toxic characteristics [5,6]. These properties make HA an ideal biomaterial for injectable hydrogels, wound patches, 3D bioprinting, tissue scaffolds, and drug delivery [7,8,9,10,11].

However, HA is currently limited in its use, owing to its relatively weak mechanical properties that prevent HA gelation into a hydrogel [8,12,13,14]. Furthermore, HA is repeatedly enzymatically degraded in a physiological environment because it is vulnerable to hyaluronidase in vivo [13,15]. This is an essential hurdle that must be overcome to further utilize HA in tissues, such as photo-crosslinking, Schiff base crosslinking, and click chemistry crosslinking, to extend the duration in vivo and obtain better mechanical properties [16,17,18]. However, the functionalized molecule has no chemical moieties with adhesive properties. These methods do not have the adhesive property needed for certain medical applications such as implant materials for surgical recovery [11,19]. Adhesive properties enable the research to be conducted in the past few years to develop an adhesive HA-derived hydrogel for medical treatment [20]. 

To achieve good adhesion and mechanical properties simultaneously, polyphenol modification has been introduced. When polyphenol is conjugated with HA, the adhesive strength increases, and polyphenol can be crosslinked. Representative materials of polyphenols with these properties include catechol, tyramine, and gallol, and they can adhere to various substrates through several interactions such as π-π stacking, hydrogen bonding, electrostatic interaction, and catechol metal correlation [21,22]. In addition, these can be oxidized under basic condition (e.g., NaOH) and treatment of NaIO_4_ or horseradish peroxidase (HRP) to crosslink via covalent bonds, or form coordination complex with metal ions [23]. However, the problem is that the catechol moieties are simultaneously involved in both crosslinking HA backbones and showing their adhesive properties. Crosslinking of phenol molecules causes change of chemical structure of phenol, so it has a possibility of losing adhesion. However, comparative analysis of cohesion and adhesion ability, according to pathway or degree of polyphenol conjugates (HA–Ca and HA–Ty), then made hydrogels using several oxidants (Figure 1). To investigate different crosslinking pathway, biological oxidant (horse radish peroxidase/hydrogen peroxide), chemical oxidant (NaIO_4_, Ammonium persulfate (APS), NaOH), and metal ions (FeCl_3_) were used. Depending on the oxidative pathway of phenol or catechol for gelation, the cohesive and adhesive strength of the HA hydrogels can be balanced because the physical amount and chemical status of these moieties involved in the crosslinking of the polymeric chains would be different. Therefore, in this study, we focused on the comparison of the cohesive and adhesive properties of crosslinked hydrogels. 

## 2. Materials and Methods

### 2.1. Materials

Sodium hyaluronate (Molecular weight = 200 kDa) was purchased from Lifecore Biomedical (Chaska, MN, USA). Dopamine hydrochloride with catechol and amine groups, tyramine hydrochloride with phenol and amine groups, N-hydroxysuccinimide (NHS), 2-(N-morpholino) ethanesulfonic acid (MES) solution (1 M), hydrochloric acid (HCl), sodium hydroxide (NaOH), horseradish peroxidase (HRP), hydrogen peroxide solution (H_2_O_2_), sodium periodate (NaIO_4_), ammonium persulfate (APS), and iron (III) chloride (FeCl_3_) were purchased from Sigma-Aldrich (St. Louis, MO, USA). 1-(3-Dimethylaminopropyl)-3-ethylcarbodiimide hydrochloride (EDC-HCl) was purchased from Tokyo Chemical Industry (Tokyo, Japan). Phosphate-buffered saline (PBS, 10X, pH 7.2) was purchased from Welgene (Gyeongsan, Korea). Sodium chloride (NaCl) was purchased from Daejung (Siheung, Korea). Anhydrous ethyl alcohol was purchased from Samchun Pure Chemical (Pyeongtaek, Korea). SpectraPor 1 Dialysis Membrane (Standard RC tubing, molecular weight cut-off (MWCO) = 6–8 kDa) was purchased from Spectrum (Rancho Dominguez, CA, USA).

### 2.2. Synthesis and Characterization of Hyaluronic Acid-Catechol and Hyaluronic Acid-Tyramine Conjugates

For the synthesis of HA–Ca, dopamine hydrochloride was conjugated with a sodium HA backbone by EDC/NHS coupling. HA (500 mg) was dissolved in 55 mL of MES buffer (0.1 M, pH 4.6). The solution was stirred for 15 min under a nitrogen atmosphere to remove dissolved oxygen capable of triggering the oxidation of catechol. After HA was fully dissolved, 190 mg/mL EDC and 115 mg/mL NHS were separately dissolved in 1 mL of MES buffer (0.1 M, pH 4.6) and then added into the reaction solution using syringes. After 10 min, 190 mg of dopamine hydrochloride dissolved in 2 mL MES buffer (0.1 M, pH 4.6) was injected into the reaction solution using a syringe. The solution was stirred for 12 h at room temperature to facilitate the reaction, with a final pH of 5.5 for prevention of further oxidation of catechol groups. To remove any unreacted free molecules, dialysis was performed using a 6–8 kDa MWCO membrane in 5 L of 100 mM NaCl solution (dissolved in acidified deionized distilled water, pH 5) for 2 days and then dialyzed in deionized distilled water for 4 h. After dialysis, the solution was lyophilized for 6 days at −80 °C, 5 mTorr. In addition, for the synthesis of HA–Ty, dopamine hydrochloride was conjugated with a sodium HA backbone by EDC/NHS coupling, following a previous report [24] and some adjustments were made. HA (500 mg) was dissolved in 50 mL of MES buffer (0.1 M, pH 5.5). After HA was fully dissolved, 20.2 mg of tyramine hydrochloride was added and stirred for 10 min. Subsequently, 190 mg/mL EDC and 237 mg/mL NHS were added together. The pH was adjusted to 4.7 with 0.1 M NaOH for optimal amide coupling reaction. After overnight reaction with constant stirring at room temperature, dialysis was performed using a 6–8 kDa MWCO membrane in 5 L of 100 mM NaCl solution (dissolved in acidified deionized water (DW), pH 5) for 2 days, dialyzed with 25% ethanol for 2 days, and then dialyzed with DW for 1 day. After dialysis, the solution was lyophilized for 6 days at −80 °C, 5 mTorr. The degree of catechol or tyramine substitution (DOS%) was analyzed by both ^1^H NMR spectroscopy (300 MHz; Varian, Palo Alto, CA, USA) and UV-vis spectroscopy (Agilent 8453; Agilent Technologies, Santa Clara, CA, USA). For obtaining ^1^H NMR spectra, each polymer was dissolved in deuterium oxide (D_2_O) at a concentration of 10 mg/mL. Additionally, for UV-vis spectra, the polymer solutions dissolved in DW were prepared at a concentration of 5 mg/mL. The absorbance at the wavelength of 280 nm (A_280_ for catechol) or 275 nm (A_275_ for tyramine) was detected. The calibration curves were established using dopamine (the concentration ranging from 15.6 μg/mL to 62.5 μg/mL) or tyramine (the concentration ranging from 3.9 μg/mL to 62.5 μg/mL). Additionally, the sample purity was confirmed by diffusion ordered spectroscopy (DOSY) (Bruker, German).

### 2.3. Preparation of Hydrogels

#### 2.3.1. HRP-Induced HA–Ca and HA–Ty Hydrogels

HRP was used as a biological oxidant for H_2_O_2_. To investigate the change in hydrogel properties based on the concentration of the oxidants, stock solutions of H_2_O_2_ (0.5 mg/mL, 1 mg/mL, and 1.5 mg/mL in pH 6 PBS for obtaining catechol (Ca): H_2_O_2_ molar ratios of 1:0.5, 1:1.0, 1:1.5, respectively) and HRP (2 unit/mL, 6 unit/mL, 18 unit/mL to make 0.1 unit/mL, 0.3 unit/mL, and 0.9 unit/mL hydrogel solution) were prepared. All hydrogels were 2 wt%.

HRP-induced HA–Ca gel was prepared as follows. HA–Ca (4 mg) was fully dissolved in 176 μL of DW, 10 μL of HRP, and 10 μL of H_2_O_2_ stock solution. After 12 h, 2 wt% HRP-induced HA–Ca gels were fabricated. HRP-induced HA–Ty gel was prepared using the same protocol, and HA–Ca was substituted with HA–Ty.

#### 2.3.2. Detection of the Free Dopamine Not Involved in HRP-Induced HA–CA Hydrogels

HRP-induced HA–Ca hydrogel (HRP 0.9 unit/mL and the molar ratio of Ca to H_2_O_2_, (1) was prepared as previous methods. The hydrogel was placed in transwell insert (24 well, 8 μm pore, Corning) and exposed to 1 mL of DW. After 12 h, the absorption spectra of the released sample solutions from the hydrogel were analyzed by UV-vis spectroscopy.

#### 2.3.3. Chemical Oxidant-Induced HA–Ca Hydrogels

For triggering oxidative crosslinking of catechols, NaOH was used as a basic additive, and NaIO_4_ and APS were utilized as the chemical oxidants. To investigate the change in gel properties based on the concentration of the oxidants, stock solutions of NaIO_4_ (0.32 mg/mL, 1.6 mg/mL, 3.2 mg/mL, 9.6 mg/mL for obtaining catechol: NaIO_4_ molar ratios of 10:1. 2:1. 1:1 and 1:3, respectively) and APS (170 mg/mL, 340 mg/mL, and 680 mg/mL to obtain molar ratios of Ca:APS molar ratios of 1:50, 1:100, and 1:200, respectively) were prepared. All hydrogels were 2 wt%.

The NaOH-induced HA–Ca hydrogel was prepared using the following steps. HA–Ca (4 mg) was fully dissolved in 190 μL of DW, and the pH was adjusted by adding 6 μL of NaOH solution. After 24 h, hydrogels were prepared. The NaIO_4_-induced HA–Ca hydrogel was prepared using the following steps. HA–Ca (4 mg) was fully dissolved in 176 μL of DW, and 20 μL of NaIO_4_ stock solution was added to fabricate NaIO_4_-induced HA–Ca gels. After 3 h, the NaIO_4_-induced HA–Ca hydrogels were prepared. The APS-induced HA–Ca hydrogel was also prepared using the same protocol by substituting NaIO_4_ with APS. The APS-induced HA–Ty hydrogel was also prepared using the same protocol by substituting HA–Ca with HA–Ty.

#### 2.3.4. FeCl_3_-Induced HA–Ca Hydrogels

To investigate the change in hydrogel properties based on the concentration of the oxidants, stock solutions of FeCl_3_ (2.0 mg/mL, 4.0 mg/mL, and 8.0 mg/mL for obtaining Ca:Fe^3+^ molar ratios of 2:1, 1:1, 1:2, respectively) were prepared. All hydrogels were 2 wt%. For gelation, HA–Ca (4 mg) was fully dissolved in 170 μL of DW. Subsequently, 20 μL of the specified concentration of FeCl_3_ solution was added and 6 μL of NaOH solution was added to adjust the pH. The gelation occurred after 18 h. 

### 2.4. Morphological Analysis and Chemical Element Mapping of HA–Ca or HA–Ty Hydrogels

To analyze cross-sectional morphology of the lyophilized HA–Ca or HA–Ty hydrogels, scanning electron microscopy (SEM; JSM7600F, Japan) equipped with an energy-dispersive X-ray spectroscopy (EDS) instrument was used. 

### 2.5. Rheological Characterization

The rheological properties of the hydrogels were determined using a Discovery Hybrid Rheometer 2 (TA Instrument, New Castle, DE, USA) with a 20 mm parallel plate geometry and a gap size of 300 μm. The storage modulus (G′) and loss modulus (G″) of the hydrogels as a function of the frequency (0.1–10 Hz) were performed at a strain of 1% at 25 °C. To test shear viscosity as a function of strain (From 0.01 to 100%), HRP-induced HA–Ty hydrogels, HRP-induced HA–Ca hydrogels, and FeCl_3_-induced HA–Ca hydrogels were performed at 25 °C. 

### 2.6. Compression Test

To compare the morphologies of the HRP-induced HA–Ca hydrogel and HA–Ty hydrogel after compression, a compression test was performed. The method for preparing the hydrogel is described in Section 2.3.1. In addition, HRP-induced HA–Ca hydrogel (2 wt%) was prepared with the concentration of 0.9 unit/mL HRP and Ca:H_2_O_2_ molar ratio of 1:1 as a final concentration. HA–Ty hydrogel (200 mg, 2 wt%) was also prepared with the same concentration of HRP and H_2_O_2_. After the preparation of the hydrogels, a weight of 700 g was placed on the gels to compress them for 10 min. After the removal of the weight from the gels, the shapes before and after compression were compared.

### 2.7. Swelling Behavior

To examine the swelling kinetics of the HA–Ca and HA–Ty hydrogels crosslinked by HRP/H_2_O_2_ catalyzed reaction (HRP 0.9 unit/mL and the molar ratio of Ca: H_2_O_2_, 1:1), each hydrogel was swollen in DW. At a pre-determined time interval (0, 0.5, 1, 2, 4, 8, 16, and 24 h), we measured the weight of each hydrogel after removal of superficial moisture. The swelling ratio (%) was calculated as the ratio of swollen weight of hydrogel to their initial weight. All experiments were triplicate.

### 2.8. Chemical Analysis of HA–Ca Crosslinking Depending on Oxidative Pathway

For investigating HA–Ca crosslinking chemistry, UV-vis spectroscopy data were collected. The method for preparing the hydrogel is described in Section 2.3.2 of this report, except for the incubation time. To obtain UV-vis spectrum data, solutions were prepared under the condition that color changes but does not form a hydrogel or color changes but before gelation time. Therefore, the NaOH-induced HA–Ca hydrogel (pH 12) was prepared with an incubation time of 4 h. NaIO_4_-induced HA–Ca hydrogel (Ca:NaIO_4_ molar ratio of 1:3) was prepared with an incubation time of 12 h. APS-induced HA–Ca hydrogel (Ca:APS molar ratio of 1:200) was prepared with an incubation time of 30 min. HRP-induced HA–Ca hydrogel (0.3 unit/mL of HRP, Ca:H_2_O_2_ molar ratio of 1:1) was prepared with an incubation time of 11 h. UV-vis spectra were recorded using an Agilent 8453 UV-vis spectrometer (Agilent Technologies, Santa Clara, CA, USA). 

### 2.9. Adhesion Strength Characterization of the Hydrogels

Tensile adhesion of the hydrogels was determined using a universal testing machine (34SC-1, Instron, IL, USA). The substrate was prepared using 30 mm × 10 mm × 0.1 mm and PET film. The samples were placed between two substrates and pressed with a weight of 1 kg for 15 min. The overlapped area was 10 mm × 10 mm, and the crosshead speed was 20 mm/min. Each sample test was repeated five times.

### 2.10. Degradation Test

To investigate degradation profile of the HA–Ca hydrogels, we prepared the hydrogels crosslinked using three different oxidation methods, such as gelation in basic condition (pH 10) and under treatment of NaIO_4_ or APS. For gelation triggered by either NaIO_4_ or APS, the molar ratio of catechol to each oxidant as 1 to 1 for NaIO_4_ and 1 to 100 for APS was utilized. After 0 (initial hydrogels) or 24 h of swelling in DW, these hydrogels were lyophilized over 12 h at −80 °C, and then the weight of dried samples was measured. Finally, the degradation (%) was calculated by the weight changes after soaking in DW compared to initial sample weight.

### 2.11. Statistical Analysis

All statistically analyzed data were determined using Student’s unpaired *t*-test. Statistically significant differences were considered when the *p*-value was less than 0.05.

## 3. Results and Discussion

### 3.1. Preparation of HA–Ca and HA–Ty Polymers and Cohesion Properties of HRP-Induced Each Hydrogel

To synthesize the desired hydrogels, modified HA was initially prepared and characterized HA was individually conjugated with catechol and tyramine to obtain HA–Ca and HA–Ty, respectively. The two modified HAs, HA–Ca and HA–Ty, were synthesized via the EDC/NHS coupling reaction (Appendix A). For HA–Ca, the DOS% of catechol was 3.7%, which was calculated by integral values of protons in aromatic rings of catechol compared to protons of HA backbone in ^1^H NMR spectra. Additionally, the DOS% analyzed by UV-vis spectra was 4.0%, like that of ^1^H NMR result. For HA–Ty, the DOS% of tyramine was 4.6% from ^1^H NMR spectroscopy and 4.5% from UV-vis spectroscopy (Figure 2a,b). When the catechol is tethered on polysaccharide, a few free catechol derivatives can be intercalated among the polymeric chains due to their intrinsic adhesiveness [24]. As shown in the results of DOSY (Appendix A), all proton signals in HA–Ty showed similar diffusion velocity (~10^−12^ m^2^/s), which indicates high purity of the polymer without free tyramine molecules (Appendix A). In contrast, a part of proton signals in HA–Ca exhibited fast diffusion behavior (~10^−10^ m^2^/s) (red dashed box, Appendix A), referring a certain degree of free dopamine entrapped in the polymer. For quantitative analysis of the free dopamine, we examined the dopamine amount not involved in gelation (e.g., HRP-triggered HA–Ca hydrogels) using UV-vis spectroscopy. As a result, 0.75% of free dopamine in total ~4 DOS% was present in the gels (Appendix A), which might be low not to significantly affect cohesive and adhesive strength of the hydrogels. Considering similar DOS% in both HA–Ca and HA–Ty, the polymers were utilized for further gelation to compare the cohesion and adhesion properties by each polyphenol (Appendix A). 

Each HA–Ca or HA–Ty hydrogel was obtained upon addition of HRP and H_2_O_2_ into the modified HA solutions, causing di–catechol and di–tyramine crosslinking in each solution (Figure 3a) [25,26]. During such crosslinking reaction, di–catechol and di–tyramine bonds form the HA polymeric network (e.g., hydrogels) with different color appearances. Before gelation, both HA–Ca and HA–Ty solutions were initially transparent; however, after gelation, the HA–Ca hydrogel had an observable reddish-brown hue, whereas HA–Ty hydrogels remained transparent (Figure 3b). This might result from the formation of di–catechol (Appendix A) [27]. In addition, for morphological analysis of each hydrogel, the cross-sectional images and chemical element mapping of the dried hydrogels were observed by SEM and EDS, respectively (Appendix A). Both hydrogels exhibited typical microporous structures with ~40 μm of pores, and all elements (e.g., carbon, nitrogen, and oxygen of each polymer) were distributed in the overall area.

Furthermore, the resulting HA–Ca hydrogels and HA–Ty hydrogels showed comparatively different storage moduli and tan δ values. HA–Ca hydrogels with different HRP concentrations (from 0.1 to 0.9 unit/mL) and different molar ratios of catechol to H_2_O_2_ (from 1:0.5 to 1:1.5) were compared to HA–Ty hydrogels prepared in the same condition. At the beginning of the study, we described a hydrogel with a tan δ ≤ 0.05 as being stiff. As regards the HA–Ca hydrogel that contained less than 0.1 unit/mL concentration of HRP, a negligible difference existed in the storage modulus and tan δ owing to its inability to form a stable hydrogel structure. By increasing the concentration, hydrogels with a concentration of over 0.3 unit/mL resulted in an HA–Ca hydrogel with biological activity. HA–Ca formed hydrogels when the concentration of HRP was 0.3 unit/mL. However, when the molar ratio of H_2_O_2_ is increased, a softer hydrogel is formed. This implies that when the concentration of H_2_O_2_ exceeds an optimum ratio, excess H_2_O_2_ will interfere with hydrogel formation rather than supporting it. 

When the HRP concentration is over 0.9 unit/mL, it provides conditions that are adequate for HA–Ca to form stiff hydrogels. A significant observation based on these measurements is that the minimum ratio of H_2_O_2_ (1:1 molar ratio and HRP 0.9 unit/mL) must be obtained to create a stiff hydrogel in HA–Ca (tan δ = 0.05 and G′ = 290 Pa at 1 Hz). Figure 3c shows that when the concentration of H_2_O_2_ was 1:0.5, even though HRP content was 0.9 unit/mL, it still failed to show stiff hydrogel formation (tan δ = 0.16 and G′ = 93 Pa at 1 Hz). Stiff hydrogels were present in HA–Ca only when the HRP concentration was 0.9 unit/mL and the H_2_O_2_ concentration was either 1:1 and 1:1.5. This reaction occurs for two reasons: (i) the larger quantity of H_2_O_2_ and HRP resulted in a higher degree of crosslinking in the hydrogel, and (ii) no H_2_O_2_ remained in the hydrogel, preventing it from interfering in the hydrogel formation (Figure 3d, Appendix A). The HA–Ca and HA–Ty hydrogels crosslinked by HRP reaction had different cohesion properties, which correspond to different crosslinking density. For details, the HA–Ty hydrogels possess much higher crosslinking ratio than that of HA–Ca, affecting their swelling kinetics [28]. To demonstrate this, we checked the swelling kinetics of each hydrogel as a function of time. As shown in Appendix A, the HA–Ca hydrogels showed higher swelling ratio (398% after 8-h incubation) than that of HA–Ty (1230% after 8-h incubation). 

To compare the shear thinning property, shear viscosity was observed based on the shear rate. Shear thinning is crucial for 3D bioprinting and injection using syringes. Both HA–Ca and HA–Ty hydrogels exhibited shear viscosities similar to those of the HA–Ty hydrogel when the shear rate is over 0.25 s^−1^. Under 0.25 s^−1^, HA–Ca showed shear thickening, whereas HA–Ty showed shear-thinning properties (Figure 3e). 

### 3.2. Adhesion Properties of HRP-Induced HA–Ca and HA–Ty Hydrogels

To verify the correlation between cohesion and adhesion strength, a versatile molar ratio of H_2_O_2_ was used to fabricate hydrogels, and the concentration of HRP was fixed at 0.3 unit/mL. We hypothesized that the stronger cohesion is induced by the greater number of phenol moieties used for crosslinking, which would decrease the adhesion properties. In previous studies, HA–Ca and HA–Ty made with a 1:0.5 H_2_O_2_ molar ratio showed a higher storage modulus (G′ = 47 Pa at HA–Ca hydrogel, G′ = 3,172 Pa at HA–Ty hydrogel) than 1:1.5 (G′ = 21 Pa at HA–Ca hydrogel, G′ = 1446 Pa at HA–Ty hydrogel) (Table 1 and Table 2). However, the result did not show significant differences because the difference in concentration of H_2_O_2_ was not significant. This shows that the structural differences between crosslinked catechol and tyramine cause differences in the adhesion strength of the hydrogel (Figure 3f). 

To obtain the adhesion strength, a lap-shear test was performed. When the concentration of H_2_O_2_ was fixed at a 1:1 molar ratio while changing the concentration of HRP, the results supported our initial hypothesis, where an increase in the degree of crosslinking in the hydrogel will result in a correlative decrease in the adhesion strength. The HA–Ca hydrogel resulted in a stronger adhesion strength at 0.3 unit/mL of HRP concentration of 0.9 unit/mL. Moreover, 0.3 unit/mL HRP-induced HA–Ca gel showed low G′ (18 Pa) compared to 0.9 unit/mL HRP-induced HA–Ca hydrogel (G′ = 290 Pa). This indicates that the cohesion becomes strong owing to more crosslinking, and the adhesion becomes weak (Figure 3g, Table 1). In addition, because of their crosslinking structure, HA–Ca and HA–Ty had different deformation shapes. When a weight of 700 g was applied to the gel for 10 min, HA–Ca showed plastic deformation, but HA–Ty was ruptured. This unexpected result might be attributed to the different structures of HA–Ca and HA–Ty being ruptured. This result can also be attributed to the different structures of HA–Ca and HA–Ty after crosslinking. This is because the HA–Ca hydrogel has one carbonyl group and two hydroxyl groups, which can exhibit adhesive properties via hydrogen bonding. Thus, HA–Ca interacts with each other through non-covalent bonding. They can lump together, even after compression (Figure 3h). These adhesion properties were supported by previous adhesion tests (Figure 3f,g).

### 3.3. Cohesion Properties of Chemical Oxidant-Induced HA–Ca Hydrogels

Previous studies have suggested several chemical oxidants, such as NaIO_4,_ and APS, which can oxidize and crosslink catechol moieties. In addition, the basic condition (e.g., NaOH) can induce di–catechol crosslinking. To investigate the effect of the oxidants, the rheological properties of each oxidizing agent were investigated. These oxidants are known to convert catechol into di–catechol (Figure 4a) [29,30]. However, the colors of the prepared hydrogels were different. The differences were verified using UV-vis spectroscopy and photographic images. HA–Ca hydrogels prepared in NaOH appeared a deep brown color with an absorption peak at 325 nm. This peak is observed when semi-quinone is generated, and it can initiate di–catechol crosslinking [31]. The hydrogel induced by APS was bright yellow and had a narrow absorption peak at 400 nm. This peak indicates the quinone form generated by the oxidation of catechol which initiates di–catechol, similar to semiquinone [32]. The hydrogel with NaIO_4_ showed a slightly yellowish hue with a broad absorption peak at 425 nm. This peak indicates the formation of di–catechol (Figure 4a,b) [33].

Variable crosslinking density can be achieved by tuning the number of oxidants. However, in the case of NaOH, the data are expressed as pH instead of the amount of NaOH. When the pH was increased from 5 to 9, HA–Ca formed a hydrogel (storage modulus > loss modulus), which indicated that the cohesive ability increased with an increase in pH. However, as the pH exceeded a certain level, the hydrogel did not form (G′ < G″ at pH 12). This weakening of the cohesive ability is likely caused by heterogeneous gelation induced by an excessively high pH (Figure 4c).

Compared to the hydrogel with NaOH, the hydrogel with APS was more elastic. The APS-added HA–Ca showed elastic hydrogel (Ca:APS was 1:100 and 1:200) when the molar ratio was above 1:50. This suggests that the cation–π interaction helps to form a stiff hydrogel. (Figure 4c,d) [34]. In addition, APS induced a transparent HA–Ty hydrogel. Tyramine is known to be crosslinked in the presence of an enzyme catalyst such as HRP, and hydrogel formation of a polymer-tyramine conjugate induced by APS has not been reported. This unexpected result also suggests that the π–cation interaction may promote the cohesive properties of hydrogels, unlike other chemical oxidants (Appendix A). The NaIO_4_ added HA–Ca hydrogel acts similar to NaOH. Furthermore, as the molar ratio of Ca:NaIO_4_ was increased from 10:1 to 1:1, the cohesive ability of the solution increased, eventually leading to the formation of the hydrogel at a 1:1 molar ratio (G′ > G″). However, as the molar ratio of NaIO_4_ is further increased, the cohesive ability decreases, which is similar to NaOH, where the weakening of its cohesive ability is caused by the aforementioned heterogeneous gelation (Figure 4e). 

Meanwhile, such chemical oxidants (e.g., NaOH) can degrade HA backbone [35,36]. According to the results, to evaluate the recovered dry weight of HA–Ca after swelling of 24 h (Appendix A), the degree of degradation was ~ 20% for NaOH-induced HA–Ca, ~ 11% for APS-induced one, and less than ~2% for the NaIO_4_-induced one. That is, a certain degree of HA backbone can be degraded by those chemical oxidants, yet it was approximately less than only 20% of total hydrogel weights.

### 3.4. Adhesion Properties of Chemical Oxidant-Induced HA–Ca Hydrogels

Because NaOH-induced hydrogels were formed at a pH of 7, the comparison of different pH conditions of the hydrogel was impossible. Therefore, adhesion strength was tested only for APS-induced hydrogels and NaIO_4_-induced hydrogels, and the stickiness according to the pH change is shown in Figure 4f. Stickiness increasingly appeared at pH 7 compared to pH 5. In the case of APS-induced hydrogels, two APS molar ratios (1:50 and 1:100) were tested. A molar ratio of 1:100 showed higher adhesion strength (3.9 ± 0.5 kPa) than that of 1:50 (0.08 ± 0.0 kPa). The reason for this result is that gel formation is difficult at a ratio of 1:50, which means the phase is almost same as liquid. This means that adhesive force did not appear if no minimal cohesive force existed (Figure 4g).

In the case of the NaIO_4_-induced HA–Ca hydrogel, the 1:0.5, and 1:1 molar ratio of Ca:NaIO_4_ did not show significant differences in adhesion strength. This is because the difference in G′ (12 Pa at 1:1 molar ratio, 18 Pa at 1:0.5, molar ratio) is not large; thus, the degree of crosslinking was not significantly different (Figure 4h). 

### 3.5. Cohesion Properties of Fe^3+^-Induced HA–Ca Hydrogels

Catechol interacts with ferric (Fe^3+^) ions via coordination bonds to form mono-, bis-, and tri-complexes. Based on the pH, this catechol-Fe^3+^ ion complexation can form mono-, bis-, and tris complexes (Figure 5a) [37]. In addition, bis and tri complexes increase cohesion strength because they can grab onto other polymers. At pH 5 and 7, the hydrogels have an observable dark green hue and a brown hue at pH 10. This brown color indicates the formation of a tri complex (Figure 5b) [38].

To tune the crosslinking density, the HA–Ca solution was crosslinked with a variable amount of Fe^3+^ ions. The molar ratio of catechol and Fe^3+^ ions was modulated (2:1, 1:1, and 1:2) while fixing the pH to 7. Among the three conditions, a 1:1 concentration is where the cohesion strength was the highest. This result might be contrary to previous reports [22] in which a ratio of 2:1 can completely form bis-complexes. A possible explanation is that the distance between catechol is too far for crosslinking between them to properly form, thus a higher concentration of mono complex instead of bis-complex will form. In the case of 1:2, the storage modulus was decreased because catechol makes more mono complexes, owing to excess Fe^3+^ ions (Figure 5c).

In addition, the cohesive ability was measured based on the change in pH when the amount of Fe^3+^ ions was at a constant 2:1 molar ratio. Hydrogel formation was observed throughout the entire pH range (pH 5–10), with the largest storage modulus (185 Pa) observed at pH 10. This is because the tri-complex that is formed exhibits a much denser crosslinking (Figure 5d) [39].

To verify the shear thinning property, the shear viscosity based on the shear rate was evaluated. From a shear rate between 0 s^−1^ and 0.6 s^−1^, shear thickening properties are observed throughout the pH range (pH 5 to 10). However, over 0.6 s^−1^, shear-thinning properties were observed at all pH ranges (Figure 5e).

### 3.6. Adhesion Properties of Fe^3+^Induced HA–Ca Hydrogels

Fe^3+^ ions improve the cohesion strength by coordinating with the hydroxyl groups of catechol. Because hydrogen bonding is the strongest interaction among molecular-molecular interactions, it was expected that the adhesion strength would drop rapidly as the hydroxyl group forms coordination bonds with Fe^3+^ ions. Therefore, it was hypothesized that the adhesive property of the hydrogel would be lowered based on the degree of crosslinking with Fe^3+^ ions. Previous data showed that the degree of crosslinking can vary based on the pH and the amount of Fe^3+^ ions. To determine the cohesive property variation based on each variable, a rheological test was performed in the two groups. The first group was used to investigate the effect of pH while fixing the molar ratio of Fe^3+^ to 2:1 (=Ca: Fe^3+^). The second group was used to investigate the effect of Fe^3+^ ions while maintaining the pH at 7. As expected, it was observed that the adhesion strength of the more crosslinked hydrogel (1:1 catechol: Fe^3+^ molar ratio) was lower than that of the less crosslinked hydrogel (2:1 molar ratio) (Figure 5f). Similar results were obtained when the pH was changed while keeping the molar ratio of FeCl_3_ constant. When pH 5 and pH 10 were compared, the storage modulus was larger at pH 10; however, the adhesion strength was smaller (1.0 kPa at pH 5 and 0.5 kPa at pH 10) (Figure 5g) and can be visibly seen in Figure 5h. The reason is that when the same amount of catechol and Fe^3+^ ions exist, the tri-complex formed can grab more catechol by forming a tri-complex. These results indicate that adhesion decreases with the degree of cohesion.

In summary, we reported the correlation between cohesion and adhesion strength based on the crosslinking pathways and crosslinking density. When the crosslinking mechanisms are compared, non-covalently bonded hydrogels obtained via metal–catechol coordination showed similar storage modulus; however, adhesion strength was lower than that of the covalently bonded hydrogel. 

Depending on the type of oxidant, the HA–Ty hydrogel crosslinked with HRP used by biological oxidants showed the highest as well as the lowest cohesion strength. In the case of HA–Ca, the radical scavenging ability was stronger than that of HA–Ca, allowing it to form less crosslinking with the biological oxidant, HRP. However, the adhesion was higher than that of the HA–Ty hydrogel. In comparison with HA–Ca, the NaIO_4_ hydrogel showed high adhesion strength and low cohesion strength. In contrast, the Fe^3+^ ion hydrogel had a fine cohesion strength; however, the adhesion strength was weak. Among the hydrogels tested, the APS hydrogel exhibited the best adhesion strength and showed a good storage modulus, similar to that of the Fe^3+^ hydrogel. In situations where both good cohesion and good adhesion are required, the APS-induced hydrogel is an option (Figure 6).

## 4. Conclusions

In conclusion, the correlation between cohesive and adhesive strength of phenol–HA hydrogels depending on their crosslinking pathway was investigated. Regarding mechanical properties of the hydrogels crosslinked by enzymatic reaction, HA–Ty formed a stiff hydrogel with high storage modulus of ~10^3^ Pa when compared to that HA–Ca. However, their adhesiveness was 21 times lower than that of HA–Ca. When the HA–Ca was covalently crosslinked by APS, the storage modulus increased up to ~10^2^ Pa, which was still lower than that of HA–Ty. That is, HA–Ty hydrogels showed strong cohesion yet weak adhesion. Among different oxidation methods (e.g., covalent or non-covalent bonds) to crosslink HA–Ca, the best option to achieve strong adhesion and cohesion was APS-triggered crosslinking. Although metal coordination network with catechol improved cohesion, their adhesive strength did not increase because most of catechols to show adhesiveness were strongly bound to metal ions. The di–catechol covalent bonds can enhance adhesion of the hydrogels due to prevention of cohesive failure. Our finding would be useful for choosing design rationale of the phenol-conjugated polymers with both robust adhesion and cohesion. 

## Figures and Tables

**Figure 1 polymers-13-03130-f001:**
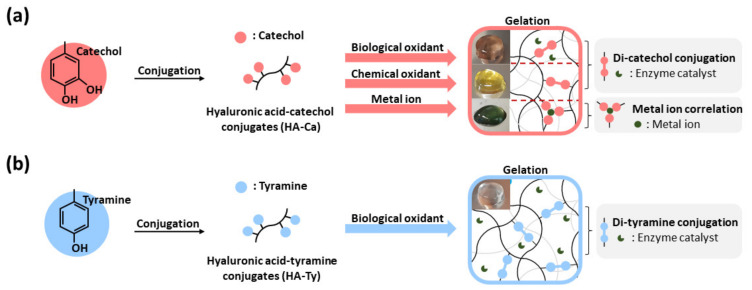
Schematic description of hyaluronic acid–polyphenol hydrogels. (**a**) Synthesis of HA–Ca conjugates (left) and their gelation by three types of oxidants (right). Biological oxidant and chemical oxidant induce crosslinking of catechol via di-catechol conjugation, and metal ion can be coordinated with catechol. (**b**) Synthesis of HA–Ty conjugates (left) and the gelation crosslinked by generating di-tyramine under a biological oxidant (right).

**Figure 2 polymers-13-03130-f002:**
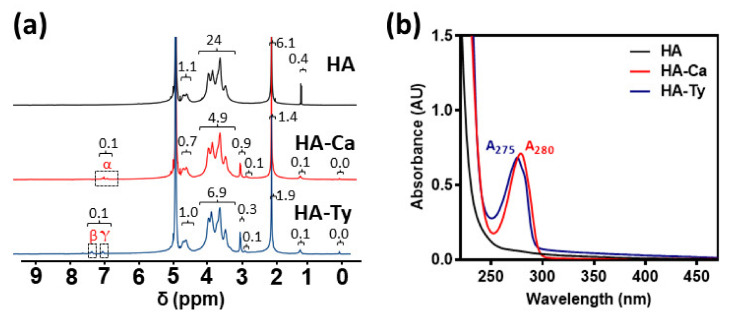
Characterization of HA–Ca and HA–Ty conjugates to evaluate degree of polyphenol substitution (%) on the HA backbone. (**a**) ^1^H NMR spectrum of HA (black), HA–Ca (yellow), and HA–Ty (blue). The ‘α’ protons indicate the protons adjacent to two hydroxyl groups in catechol moieties. The ‘β and γ’ protons indicate the protons adjacent to hydroxyl group in tyramine moieties. (**b**) UV–vis spectra of HA (black), HA–Ca (red), and HA–Ty (blue) solutions. The absorbance at the wavelength of 280 nm (A_280_) means the presence of catechol, and the absorbance at 275 nm (A_275_) indicates tyramine.

**Figure 3 polymers-13-03130-f003:**
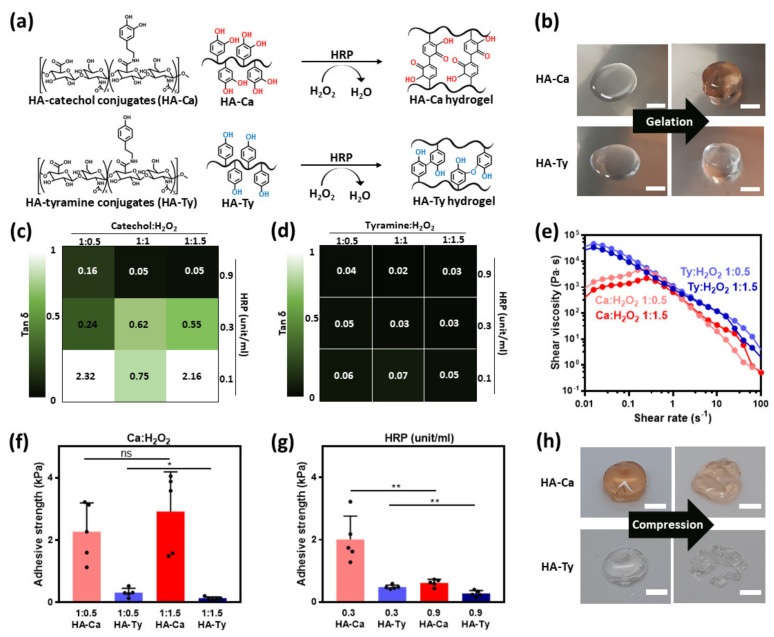
Comparative observation of HA–Ca and HA–Ty hydrogel which are crosslinked by biological oxidants. (**a**) HA–Ca (top) and HA–Ty (bottom) are crosslinked by horseradish peroxidase (HRP) and hydrogen peroxide (H_2_O_2_) to form HA–Ca and HA–Ty hydrogel, respectively. (**b**) Gelation of HA–Ca solution (top) and HA–Ty solution (bottom) before (left) and after gelation (right) (scale bar = 5 mm). Rheological properties of (**c**) HA–Ca and (**d**) HA–Ty hydrogels with different HRP and H_2_O_2_ concentrations. (**e**) Shear viscosity of HA–Ca (reddish) and HA–Ty (bluish) as a function of shear rate in different ratios of H_2_O_2_ showing a shear-thinning property. Adhesion strength of HA–Ca and HA–Ty hydrogels in (**f**) different H_2_O_2_ molar ratio at 0.3 unit/mL of HRP concentration and (**g**) different concentrations of HRP at H_2_O_2_ molar ratio of 1:1 (*n* = 5, mean ± SD) (* *p* < 0.05, ** *p* < 0.01, ns = not significant). (**h**) Images of HA–Ca hydrogel (top) and HA–Ty hydrogel (bottom) before (left) and after (right) compression (scale bar = 5 mm).

**Figure 4 polymers-13-03130-f004:**
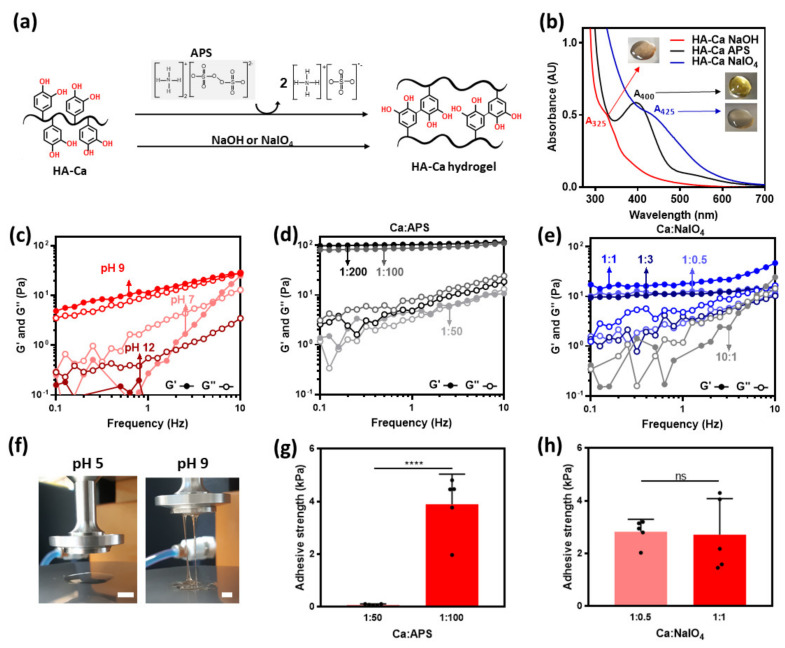
Adhesive and cohesive properties of chemical oxidants-induced hydrogels. (**a**) Catechol forms di-catechol covalent bonds owing to APS, NaIO_4_, NaOH. (**b**) UV-vis spectra of chemical crosslinking-induced HA–Ca solutions by three different oxidants. Frequency sweep-storage (G′) and loss (G″) moduli of HA–Ca hydrogels at 1% strain, oxidized with (**c**) NaOH, (**d**) NaIO_4_, and (**e**) APS with different molar ratios of catechol and the oxidant. (**f**) Images of NaOH-induced hydrogel with different pH (scale bar = 5 mm). Adhesion strength of various oxidants-induced HA–Ca hydrogels on PET substrate in (**g**) different Ca:APS molar ratio and (**h**) different Ca:NaIO_4_ molar ratio (*n* = 5, mean ± SD) (**** *p* < 0.0001, ns = not significant).

**Figure 5 polymers-13-03130-f005:**
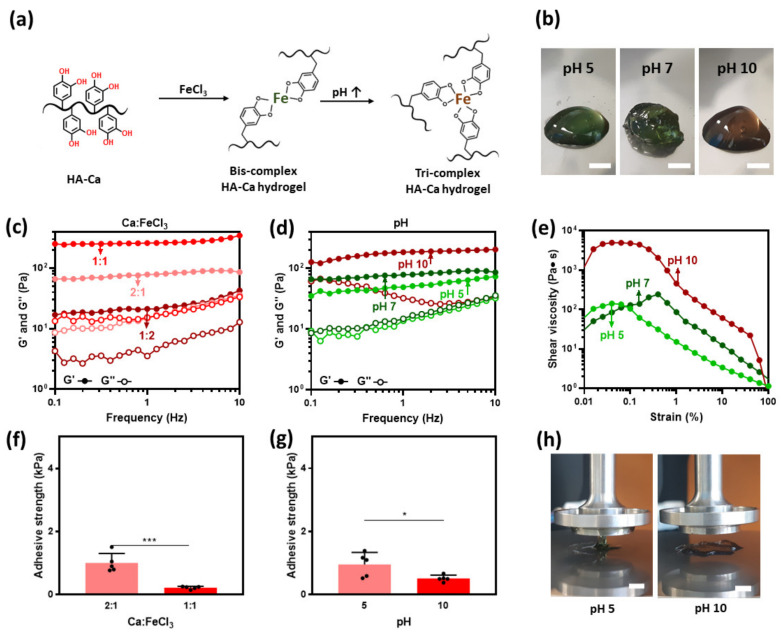
Adhesive and cohesive properties of FeCl_3_-induced hydrogels. (**a**) Catechol forms a non-covalent coordination bond owing to Fe^3+^ ion. (**b**) Photos of FeCl_3_-induced HA–Ca hydrogel prepared at different pH (5, 7, and 10) (scale bar = 5 mm). (**c**) in different pH at Ca:Fe^3+^ molar ratio of 2:1 and (**d**) in different Fe^3+^ molar ratios at pH 7. (**e**) Shear-thinning properties of the hydrogel in different pH at Ca:Fe^3+^ molar ratio of 2:1. Adhesion strength of the hydrogels in (**f**) different catechol:Fe^3+^ molar ratios at pH 7, (**g**) different pH at Ca:Fe^3+^ molar ratio of 2:1 (*n* = 5, mean ± SD) (* *p* < 0.05, *** *p* < 0.001), and (**h**) their images at pH 5 (left) and pH 10 (right) (scale bar = 5 mm).

**Figure 6 polymers-13-03130-f006:**
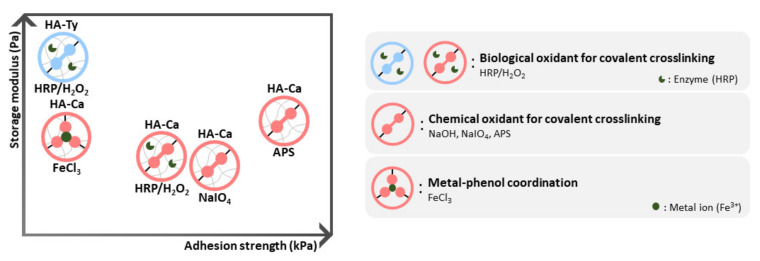
Correlation of adhesion and cohesion strength varying by crosslinking mechanisms. Hydrogels are indicated in the graph (left) based on the level of adhesion strength and storage modulus.

**Table 1 polymers-13-03130-t001:** Storage modulus (G′) of HA–Ca hydrogels at different HRP concentrations and the stoichiometric ratio of H_2_O_2_ to 0.5, 1, or 1.5.

G′ (Pa)	H_2_O_2_ 1:0.5	H_2_O_2_ 1:1	H_2_O_2_ 1:1.5
HRP 0.3 unit/mL	93	290	205
HRP 0.9 unit/mL	47	18	21

**Table 2 polymers-13-03130-t002:** Storage modulus (G′) of HA–Ty hydrogels at different HRP concentrations and the stoichiometric ratio of H_2_O_2_ to 0.5, 1, or 1.5.

G′ (Pa)	H_2_O_2_ 1:0.5	H_2_O_2_ 1:1	H_2_O_2_ 1:1.5
HRP 0.3 unit/mL	5146	4736	4305
HRP 0.9 unit/mL	3172	1551	1446

## Data Availability

The data presented in this study are available in the article.

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
