# Peer review of "Phenol–Hyaluronic Acid Conjugates: Correlation of Oxidative Crosslinking Pathway and Adhesiveness"

_polymers, 2021, doi:10.3390/polym13183130_

Round 1

Reviewer 1 Report

In general, the work is not clearly  presented. The language is poor and the tables are overcrowded with information.  The authors  should move the 1H NMR from the supplementary part to the main test. Also, the figures in the supplementary part were not mentioned in the  manuscript.  DOSY NMR should be added to see whether the product is pure. I also suggest adding all the integrals of 1H NMR not only the aromatic signals.

The English language should be revised. For example, the conclusions should be written in passive voice. Not only we observed, we compared...

Line 100 all the molecules are tiny please use for scientific language

line 169 After 18 h, FeCl3-induced HA-Ca hydrogels were prepared...check the language 

Figure 2 c and d are not clear. It is better if the authors create a table concentration versus moduli because the figure is overcrowded and confusing.

In general, the authors should describe whether they have observed degradation because the oxidants are known to degrade HA- backbone. The straightaway method is to determine swelling degree, dry the materials and compare the amount of HA recovered.  NaIO4 is generally used for oxidation of HA, . NaOH is not considered as oxidant, but a base .Please correct based on the mechanism of crosslinking. 

Line 267 In addition, because of their crosslinking structure...correct the language. 

The literature should be changed and more relevant references should be used. For example, dermal fillers based on HA use divinylsulfone or BDPE, which are different to tyramine mechanism of crosslinking. 

Many references related to chitosan and alginate are not related to this topic. While important references dealing with the chemistry of hyaluronan were ignored

Cross-Linking Chemistry of Tyramine-Modified Hyaluronan Hydrogels Alters Mesenchymal Stem Cell Early Attachment and Behavior | Biomacromolecules (acs.org)

Abu-Hakmeh, A., Kung, A., Mintz, B.R. et al. Sequential gelation of tyramine-substituted hyaluronic acid hydrogels enhances mechanical integrity and cell viability. Med Biol Eng Comput 54, 1893–1902 (2016). https://doi.org/10.1007/s11517-016-1474-0

Hong, B., Park, S. & Park, W. Effect of photoinitiator on chain degradation of hyaluronic acid. Biomater Res 23, 21 (2019). https://doi.org/10.1186/s40824-019-0170-1

Please lines 67-68, define better the novelty of this work with a good hypothesis. 

Reviewer 2 Report

           The manuscript ID polymers-1363518, entitled “Phenol-hyaluronic acid conjugates: Correlation of oxidative crosslinking pathway and adhesiveness" by authors: Jungwoo Kim, Sumin Kim, Donghee Son, Mikyung Shin, is interesting, well written and informative. The authors have investigated the correlation of cohesion strength and adhesion strength of hyaluronic acid-catechol (HA-Ca) and hyaluronic acid-tyramine (HA-Ty) conjugates based on different cross-linkers and oxidants, as well as on different molar ratios. As a result, HA-Ty formed a stiff hydrogel with excellent storage modulus, but with very low adhesion strength, compared to HA-Ca. Covalent bonding showed high adhesion strength in all types of hydrogels except HA-Ty, compared to non-covalent bonding. The horseradish peroxidase-induced HA-Ty hydrogel showed the highest storage modulus. The ammonium persulfate-induced HA-Ca hydrogel had both good cohesion and adhesion properties.

The abstract provides a good overview of the manuscript.

Introduction gives the short description of the state of art, as well as importance of application hyaluronic acid as a biomaterial.

Methods part provides necessary data about different applied methods.

In the Results part authors have analyzed and presented obtained results, in correlation to previous investigations.

In the Conclusion,  the possible future characterization and application of obtained and selected product should be added.

Authors cited 35 relevant papers in the manuscript.

After careful reading, in my opinion, this manuscript needs some additional improvements according specific comments to get accepted for publication as follow:

  • In the part1. Materials, please, it is needed to insert necessary data for Catechol.
  • It will be more precise to add data about Hyaluronic Acid-Tyramine rewrite caption e.g.: "2. Synthesis and Characterization of Hyaluronic Acid-Catechol and Hyaluronic Acid-Tyramine Conjugates"
  • Also, it would be better to emphasize differences between the process for synthesis of HA-Ty in relation to the synthesis of HA-Ca, especially lines 113-115 and 98-101.
  • Is the lyophilization process the same for HA-Ty as for HA-Ca (for 6 days)? Could authors provide the applied stages in the lyophilization conditions process (pressure, temp. and their duration)?
  • For the characterization of the HA-Ca and HA-Ty conjugates, author have to apply the SEM, TEM, and/or EDS methods for surface morphology analysis, as well as to investigate swelling behavior.
  • Lines 166-167: Please, check and correct duplicated terms.
  • Line 204: It will be better to rename part 3. Results as: Results and Discussion
  • Lines 300-301: Please, it is needed to check and correct Figure 2 caption, for parts:
    • "(a) HA-Ty (top) and HA-Ca (bottom)", should be: "(a) HA-Ca (top) and HA-Ty (bottom); and
    • "(c) HA-Ty and (d) HA-Ca hydrogels..." should be: "(c) HA-Ca and (d) HA-Ty hydrogels...".
  • Lines 400-401: Part (b) in Figure 4 caption was missed. Please, it is needed to insert description.
  • Please, it is needed to correct caption of Supplementary Figure S1: description for (b) part was missed: probably, it would be better: "Synthesis and characterization of hyaluronic acid (HA)-phenol conjugates. (a) hyaluronic acid-catechol conjugate (HA-Ca) and (b) hyaluronic acid-tyramine conjugate (HA-Ty) synthesis using carbodiimide coupling reaction..."
  • Please, avoid 1st person plural  and rewrite all sentences to 3rd person plural, because it is common for scientific papers to be written in passive.

After revision by the authors, this manuscript can be considered for publishing in the Polymers journal.

Best regards,

Reviewer

Round 2

Reviewer 1 Report

There are one or two small problems with English. However, the authors have done an excellent correction in the science. As they are not native speakers and the manuscript is clear. I do not have any problem with acceptance in the current form. 

As example line 905, our findings.

Line 906 instead of polymer I would add hyaluronan

I saw two or three small corrections, but they are not significant for the quality of the work.